# Wood Cellulose Nanofibers Grafted with Poly(ε-caprolactone) Catalyzed by ZnEu-MOF for Functionalization and Surface Modification of PCL Films

**DOI:** 10.3390/nano13131904

**Published:** 2023-06-21

**Authors:** Jinying Pang, Tanlin Jiang, Zhilin Ke, Yu Xiao, Weizhou Li, Shuhua Zhang, Penghu Guo

**Affiliations:** 1Guangxi Key Laboratory of Electrochemical and Magnetochemical Functional Materials, College of Materials Science and Engineering, Guilin University of Technology, Guilin 541004, Chinakzl@glut.edu.cn (Z.K.); 2Guangxi Key Laboratory of Natural Polymer Chemistry and Physics, College of Chemistry and Materials, Nanning Normal University, Nanning 530001, China; 3College of Resources, Environment and Materials, Guangxi University, Nanning 530004, China; 4Key Laboratory of Petrochemical Pollution Control of Guangdong Higher Education Institutes, Guangdong Provincial Key Laboratory of Petrochemical Pollution Process and Control (College of Chemistry), Guangdong University of Petrochemical Technology, Maoming 525000, China; 5School of Materials Science and Engineering, Xiamen University of Technology, Xiamen 361024, China

**Keywords:** ZnEu-MOF, catalyst, GCL, fluorescence, light composite films, mechanical properties

## Abstract

Renewable cellulose nanofiber (CNF)-reinforced biodegradable polymers (such as polycaprolactone (PCL)) are used in agriculture, food packaging, and sustained drug release. However, the interfacial incompatibility between hydrophilic CNFs and hydrophobic PCL has limited further application as high-performance biomaterials. In this work, using a novel **ZnEu-MOF** as the catalyst, graft copolymers (GCL) with CNFs were grafted with poly(ε-caprolactone) (ε-CL) via homogeneous ring-opening polymerization (ROP), and used as strengthening/toughening nanofillers for PCL to fabricate light composite films (LCFs). The results showed that the **ZnEu-MOF** ([ZnEu(L)_2_(HL)(H_2_O)_0.39_(CH_3_OH)_0.61_]·H_2_O, H_2_L is 5-(1H-imidazol-1-yl)-1,3-benzenedicarboxylic acids) was an efficient catalyst, with low toxicity, good stability, and fluorescence emissions, and the GCL could efficiently promote the dispersion of CNFs and improve the compatibility of the CNFs and PCL. Due to the synergistic effect of the ZnEu-MOF and CNFs, considerable improvements in the mechanical properties and high-intensity fluorescence were obtained in the LCFs. The 4 wt% GCL provided the LCF with the highest strength and elastic modulus, which increased by 247.75% and 109.94% compared to CNF/PCL, respectively, showing the best elongation at break of 917%, which was 33-fold higher than CNF/PCL. Therefore, the **ZnEu-MOF** represented a novel bifunctional material for ROP reactions and offered a promising modification strategy for preparing high-performance polymer composites for agriculture and biomedical applications.

## 1. Introduction

Cellulose nanofibers (CNFs) are attractive nanofillers for the modification of biodegradable polymers due to their high aspect ratio, low density, flexibility with mechanical strength [1,2], and environmental friendliness [1,3,4]. CNFs can improve the mechanical and thermal properties of polymer composites, especially in biodegradable polymers [4,5,6]. Polycaprolactone (PCL), one of the most promising biodegradable polymers, exhibits good biocompatibility, excellent biodegradability, and can be utilized for plant growth and sustainable agriculture [7,8], drug carriers [9,10], and food packaging [11]. However, PCL has not yet been produced from renewable resources, and its low mechanical strength, strong hydrophobicity, and poor cell affinity limit its further application. Therefore, it is necessary to improve the biodegradable components of PCL-based materials. From this perspective, CNF-reinforced PCL can provide biodegradability, low cost, and increase the content of bio-based composites. Compared to cellulose nanocrystals (CNCs) and wood pulp fibers, CNFs are more suitable for reinforcing PCL [3]. Due to interfacial incompatibility between hydrophilic CNFs and hydrophobic PCL, the enhancement will not be ideal [12]. In addition, CNFs will easily agglomerate, making them difficult to uniformly disperse in PCL, and this aggregation results in weak reinforcement CNFs in PCL. To resolve this problem, polymer grafting [13,14], sulfonation [15,16], and acetylation [17] have been used to modify CNFs in numerous studies, with polymer grafting considered the most effective strategy among the above modification methods [18,19]. Stannous caprylate and 4-dimethylaminopyridine (DMAP) are commonly used catalysts in the market for the preparation of nanocellulose-grafted PCL [20,21]. However, the catalysts used for the grafting of polymers are highly toxic with low catalytic efficiency [21,22]. Furthermore, non-functional CNFs cannot provide PCL with the fluorescence properties required for numerous applications. The side chain grafting of PCL on CNFs can solve the physical entanglement formed by CNFs at the phase interface, and enhance the interfacial compatibility between them, thus, obtaining the required mechanical properties of PCL and improving its hydrophobicity. However, only a small number of PCL side chains can be grafted onto CNFs, with minimal reported improvement of the polymer’s mechanical properties [18]. Grafted PCL polymers onto CNCs via ROP, the elastic modulus increased, but the tensile strength and strain at break of grafting copolymers/PCL composites decreased compared to pure PCL [23]. PCL could be grafted from holocellulose wood fibers (HC) by ROP, the elastic modulus, strength, and elongation at break of the PCL composites improved by almost 60% and 67%, respectively [12]. The result studies showed that when the grafting copolymers added to the PCL, the modulus and tensile strength of the composites were increased, but the elongation at break decreased [24]. Therefore, according to the above literature, although the mechanical properties of the composites could be improved, the catalysts used for grafting the polymer were highly toxic, with low catalytic efficiency, and they could not provide PCL with fluorescence properties. Hence, it is essential to develop green and highly efficient catalysts for the GCL reaction, and to design fluorescent additives that can appropriately bind to the film matrix through the intermediate support material.

Because of their structural and compositional tunability, metal-organic frameworks (MOFs) with organic-inorganic hybrid compositions can serve as promising materials for various applications such as heterogeneous catalysis, fluorescence, adsorption, and controlled release [10,25,26,27,28,29,30,31]. In this preliminary work, highly luminescent PCL polymer films incorporated with a europium (III) complex Eu-MOF were studied [32,33]. Among the reported structures, mixed-metal MOFs possessing properties of metals and beyond could further improve the performances of MOFs. For example, well-documented MIL-53(Cr/Fe), Fe^III^_2_M^II^-MIL-100 (M = Co, Ni or Mg), MOF-74 (Mg/Ni, Mg/Co), UiO-66 (Zr/Ti), HKUST-1 (Cu/Fe, Cu/Co, or Cu/Pd), and ZIF-67 (Zn/Co), etc. [34,35,36,37,38,39]. Zn-contained complexes can be used as catalysts for the graft co-polymerization reaction between PCL and nanocellulose, and Eu-based complexes are well-known highly efficient fluorescent materials [40,41]. To reduce the toxicity of the catalyst in the GCL reaction, an organic linker containing oxygen and nitrogen coordination sites simultaneously that could coordinate with both the Zn and Eu ions was needed to produce mixed-metal MOFs with dual-functional metal centers. A preliminary assessment showed that amphiphilic polymers were not only non-toxic to cells, but also promoted cell growth within a certain concentration range [22,41].

In this research, 5-(1H-imidazol-1-yl)-1,3-benzenedicarboxylic acid was deliberatively employed as an organic linker, along with Zn and Eu as the bimetallic nodes, to construct porous mixed-metal MOF, namely **ZnEu-MOF**. Multiple techniques including single-crystal X-ray diffraction (XRD), powder XRD, ^1^H NMR (proton nuclear magnetic resonance), FT-IR (Fourier transform infrared), thermogravimetric (TG) analysis, and photoluminescence were utilized to investigate its properties. GCL was characterized by ^1^H NMR, FT-IR, and TG, and the reaction mechanism of the catalysts was studied. In addition, light composite films were prepared by blending the GCL with PCL, and the macroscopic mechanical properties and morphology properties, physical-chemical properties, fluorescence and its mechanism, and the in vitro cytotoxicity of LCF were studied.

## 2. Experiments and Methods

### 2.1. Materials

All chemicals and solvents were commercially purchased and used without further purification. 5-(1H-imidazol-1-yl)-1,3-Benzenedicarboxylic acid (98%) was purchased from the Shanghai Bide Pharmaceutical Technology Company, and zinc chloride hexahydrate (98%) was obtained from the Shanghai Macklin Biochemical Technology Company. N, N-dimethylformamide (DMF), dichloromethane, anhydrous ethanol, and acetonitrile (MeCN) were purchased XiLong Chemical Company, and europium nitrate hexahydrate was purchased from the Zhengzhou Alpha Chemical Company. Ionic liquid ([Bmim]Cl) and ε-CL were obtained from the Lanzhou Institute of Chemical Physics. Nanocellulose and ε-caprolactone were purchased from the Institute of Chinese Academy of Sciences and the Wuhan Huaxiang Kejie Biotechnology Company, respectively. PCL was purchased from Shenzhen Guang Hua Wei ye Co., Ltd. Ultrapure water was prepared in the laboratory. The HEK293T cell line was obtained from the American Type Culture Collection (ATCC).

### 2.2. Synthesis of the ZnEu-MOF

A mixture of 5-(1H-imidazol-1-yl)-1,3-benzenedicarboxylic acid (0.40 mmol, 0.093 g), Eu(NO_3_)_3_∙6H_2_O (0.10 mmol, 0.045 g), ZnCl_2_∙6H_2_O (0.1 mmol, 0.024 g), acetonitrile (4 mL), N, N-dimethylformamide (4 mL), and methanol (2 mL) was placed in a 15 mL Teflon-lined solvothermal synthesis autoclave, and the pH value of the mixture solution was adjusted to approximately 6 by 85 wt% formic acid solution under continuous stirring for 30 min. After heating the oven to 120 °C for 3 days, colorless flake crystals of ZnEu-MOF were obtained by cooling the reactor to 25 °C (yield of 0.0778 g, ca. 82.4% based on Zn^2+^). The crystals were heated in a tube furnace at 200 °C for 14 h under vacuum (0.09 MPa). Anal. Calc. for ZnEu-MOF: C_33.61_H_24.22_EuN_6_O_14_Zn (*Mr* = 953.52), calc.: C, 42.33; H, 2.56; N, 8.81%; found: C, 42.24; H, 2.62; N, 8.85%. The following FT-IR data were obtained for ZnEu-MOF (KBr, cm^−1^, Figure 1i): 3380 (s), 3144 (m), 2931 (w), 1629 (s), 1593 (s), 1560 (s), 1521 (m), 1438 (m), 1385 (s), 1290 (w), 1209 (w), 1143 (w), 1090 (w), 785 (m), 752 (m), 715 (m), 673 (m), and 480 (w).

### 2.3. Synthesis of GCL

The GCL samples were synthesized according to our previous work [22,41] and Figure 2 illustrates the synthesis route for GCL. The detailed processes are described in Appendix A.

### 2.4. Preparation of PCL and the Composite Films

The mass fraction values of GCL in the LCFs were 2 wt%, 4 wt%, 6 wt%, 8 wt%, and 10 wt%. According to our previous work [42], the GCL was directly mixed with a dichloromethane solution of PCL, based on the solid content, with the mixed solution dispersed by an ultrasonic cell disrupter, which was then slowly poured into a 15-cm diameter culture dish. The GCL/PCL composite film with a thickness of approximately 0.2 mm was fabricated when dichloromethane was fully evaporated.

### 2.5. Characterization

The single-crystal X-ray diffraction data of **ZnEu-MOF** were collected by an APEX-II CCD diffractometer (Germany) with graphite mono-chromated *Mo-Kα* radiation (*λ* = 0.71073 Å) at 20 ± 1 °C in *ω* scan mode. The raw frame data were integrated with the SAINT program, and the structure was solved by direct methods using SHELXT [43] and refined by the full-matrix least-squares on F^2^ using SHELXL-2018 [44] within the OLEX-2 GUI [45]. An empirical absorption correction was applied with spherical harmonics, implemented in the SCALE3 ABSPACK scaling algorithm, and all non-hydrogen atoms were refined anisotropically. All hydrogen atoms were positioned geometrically and refined as riding. The calculations and graphics were performed by SHELXTL [43,44]. The FT-IR spectra were recorded by a Nicolect Nexus 470 FT-IR infrared spectrometer from 4000 to 400 cm^−1^ using KBr pellets. The fluorescence spectra were recorded by an F-4600 fluorescence spectrophotometer setup, where the slit width was 5 nm and the test voltage was 700 V. The samples were tested on Q20 (TA Instrument, New Castle, Delaware, United States) DSC equipment. Approximately 5 mg of the samples were conducted under nitrogen protection, with a heating rate of 10 °C/min. The samples were heated from 40 °C to 80 °C and cooled to 0 °C, and the temperature of the second heated cycle ranged from 0 °C to 80 °C. The crystallinity (*X_c_*) of the LCFs was calculated by Equation (1):*X_c_* = Δ*H_m_*/((1 − *ω*) × Δ*H_m_^θ^*),(1)
where Δ*H_m_* is the experimental melting enthalpy of the sample, Δ*H_m_^θ^* is the melting heat for 100% of polymer crystallization (136 J/g) [42], and *ω* is the mass fraction of the GCL.

Powder XRD measurements were performed on a Rigaku Ultima IV powder diffractometer (Japan) from 5° to 50°. The TG data were obtained by a NETZSCH TG209 instrument, which was set from room temperature to 700 °C under nitrogen conditions with a heating speed of 10 °C/min.

The scanning electron microscope (SEM) images were obtained by a field-emission TESCAN MIRA LMS. The tensile properties of the PCL and composite films were obtained by a Shimadzu AGS-X10KN universal testing machine at a tensile rate of 200 mm/min. The film materials were cut into standard dumbbell shapes with a length and width of 80 and 4 mm, respectively. The contact angles of the composite films were tested by a JC2000D machine (Shanghai Zhongchen Digital Technology Equipment Co., Ltd., shanghai, China), and the TD-DFT method was used to calculate the excited states and obtain the ultraviolet spectra, as theoretically predicted. The functional selected of ZnEu-MOF for structure optimization, frequency calculation, and excited state calculation was PBE0, with a basis set of 6–311 g(d) used for non-metallic atoms, and the Stuttgart pseudopotential group (SDD) was used for the transition metal zinc. For europium, a rare earth element with an obvious relativistic effect, the pseudopotential group MWB52 was used, which was labeled by the large Stuttgart pseudopotential potential. Finally, the results were analyzed by Multiwfn3.7, and the molecular frontier orbital information of the ZnEu-MOF molecules was obtained to analyze the fluorescence mechanism.

### 2.6. In Vitro Cytotoxicity

The cytotoxicities of GCL-D/PCL and 4 wt% LCF were investigated by using HEK293T cells. The detailed processes are described in Appendix A.

### 2.7. Computational methods

Density functional theory (DFT) calculations were performed using Gaussian 09. The molecular structures of ZnEu-MOF was fully optimized by PBE0 functional, with a basis set of 6–311 g(d) used for non-metallic atoms [41]. The TD-DFT method was used to calculate the excited states and obtain the ultra-violet spectra, as theoretically predicted. The PBE0 functional was used for excited state calculation, the Stuttgart pseudopotential group (SDD) was used for the transition metal zinc. For europium, a rare earth element with an obvious relativistic effect, the pseudopotential basis set MWB52 was used, which was labeled by the large Stuttgart pseudopotential potential [46]. Finally, the results were analyzed by Multiwfn3.7 [47], and the molecular frontier orbital information of the ZnEu-MOF molecules was obtained to analyze the fluorescence mechanism.

## 3. Result and Discussion

### 3.1. Characterization of **ZnEu-MOF**

Single-crystal XRD analysis showed that **ZnEu-MOF** belonged to the monoclinic system with space group *C*2/*c*. (Appendix A). As shown in Figure 1a–d, the smallest crystallographic unit contained one Eu (III) ion, one Zn (II) ion, two L ligands, one HL ligand, a 0.39 coordinated water and 0.61 methanol molecule to Eu^3+^, and one free lattice water.

The Zn1 atom was coordinated with three O atoms from three different L^2−^ molecules [Zn1–O2 = 1.924(5) Å; Zn1–O7 = 1.929(4) Å; Zn1–O11 = 1.933(4) Å] and one N atom from another L^2−^ molecule [Zn1–N6^c^ = 2.022(7) Å; Zn1–N6′^c^ = 2.00(4) Å, symmetry code: (c) 1 − *x*, 1 − *y*, 1 − *z*] (Figure 1a,b and Appendix A). The coordination configuration of Zn1 possessed tetrahedral coordination geometry (label, T-4; symmetry: *T*_d_) using SHAPE 2.1 (Appendix A). The Eu1 atom displayed spherical capped square antiprism coordination geometry (label, CSAPR-9, Appendix A) coordinated with O1, O10, O6, O8^a^, O9^a^, O12^b^, O13^b^, and N4^d^(N4’^d^), symmetry codes: (a) −*x*, *y*, 0.5 − *z;* (b) 0.5 − *x*, 0.5 − *y*, 1 − *z;* (d) *x*, 1 − *y*, 0.5 + *z*) from six L^2−^ ligands and O5(O5′) from the solvent molecules (the ratio of o5 to o5’ was 39:61, so the sum of their occupancy was 1). Of note, the H_2_L ligand had three coordination models (Figure 1e–g): The first coordination mode of L^2−^ was *μ*^4^-*η*^1^:*η*^1^:*η*^1^:*η*^1^:*η*^1^-L^2−^(2Eu2Zn) (Figure 1e), the second coordination mode of L^2−^ was *μ*^4^-*η*^1^:*η*^1^:*η*^1^:*η*^1^:*η*^1^-L^2−^(3Eu1Zn) (Figure 1f), and the third coordination mode of HL^−^ was *μ*^2^-*η*^1^:*η*^1^-L^2−^(1Eu1Zn) (Figure 1g). The **ZnEu-MOF** constructed the 3D micropore network through the *μ*^4^-*η*^1^:*η*^1^:*η*^1^:*η*^1^:*η*^1^-L^2−^(3Eu1Zn) and *μ*^4^-*η*^1^:*η*^1^:*η*^1^:*η*^1^:*η*^1^-L^2−^(2Eu2Zn) bridge ligands (Figure 1c,d). The 3D micropore network was further stabilized by abundant hydrogens (O4-H4···O13^v^, 2.663 (8) Å, O5-H5···O12^vi^, 2.705 (7) Å, O5-H5B···O9^vii^, 2.705 (7) Å; symmetry code: (v) *x* − 0.5, *y* + 0.5, *z*; (vi) *x* − 0.5, 0.5 − *y*, 1 − *z;* (vii) −*x*, *y*, 0.5 − *z*). According to Platon calculations, each unit cell (10991.3 Å^3^) of the **ZnEu-MOF** had a pore volume of 4698.8 Å^3^ and porosity ratio of 42.8% removed from the solvent molecules. The crystallographic details are provided in Appendix A, and the selected bond lengths and angles for **Zn-MOF** are listed in Appendix A.

The powder XRD results of the **ZnEu-MOF** are plotted in Figure 1h. The results matched well with the simulated pattern of the **ZnEu-MOF**, indicating the high purity of the obtained **ZnEu-MOF** sample.

According to the analysis of the infrared spectrum data in Figure 1i, the broad absorption peak at approximately 3380 cm^−1^ was generated by the stretching vibrations of −OH from the water molecules in the crystal lattice [48]. Peaks near 3144 cm^−1^ originated from the C-H stretching vibration of the benzene ring. The typical stretching vibration of the C=O double bond of ligand H_2_L was observed at approximately 1720 cm^−1^, while in the ZnEu-MOF spectrum, the corresponding peak red-shifted to 1629 cm^−1^, indicating that the carboxyl group in the ZnEu-MOF was deprotonated and coordinated with the metal ions [49]. The characteristic asymmetric stretching vibration of the C=C bond in the benzene ring was observed at 1450–1610 cm^−1^ [50],and the vibrations of the C=N double bond of the imidazole ring in the **ZnEu-MOF** and H_2_L ligand were located at 1593 cm^−1^ and 1612 cm^−1^, respectively [51]. In addition, the characteristic coordination peak of the Zn-O bond in the **ZnEu-MOF** appeared at 480 cm^−1^ [52]. Therefore, the infrared spectrum of the **ZnEu-MOF** sample was consistent with its structure, as determined by the single-crystal XRD data.

A TG test was performed for the **ZnEu-MOF**, with the results shown in Figure 1j, the power XRD of TG final residual material with **ZnEu-MOF** is showed in Appendix A. The first thermal decomposition stage of the complex was in the temperature range of 25–105 °C. The TG curve of the **ZnEu-MOF** indicated weight loss, and the corresponding DTG curve showed an endothermic peak. Calculated by the weight loss (4.9%), it was equivalent that ZnEu-MOF lost 0.69 coordination water molecules and 0.31 coordination methanol molecules, along with one lattice water molecule (calculated value of 4.8%). The above results were consistent with the composition analysis results of the ZnEu-MOF. The [ZnEu(L)_2_(HL)] portion of the **ZnEu-MOF** remained stable up to 156 °C. The second decomposition stage of the complex started at 156 °C, and the decomposition temperature ranged from 156 °C to 700 °C. The remaining material (approximately 50% weight ratio) was a mixture of zinc and europium oxide [22].

### 3.2. Characterization of GCL and Catalytic Synthesis Mechanism

In our previous work [22,41], we reported on the similar characterization and synthesis mechanism of GCL. The 1H NMR spectrum of GCL is shown in Appendix A, where the grafting ratio of GCL was 79.6%(it was calculated-based on Appendix A), which was catalyzed by the **ZnEu-MOF**. The **ZnEu-MOF** had a superior catalytic effect compared to other literature-reported catalysts (Table 1). The FT-IR spectra of the CNF and GCL samples are shown in Appendix A, the TGA results of CNF and GCL are shown in Appendix A. The analysis is provided in Appendix A.

The main link in the process of CNF grafted with ε-caprolactone was to trigger the ring-opening of ε-caprolactone to form a long chain of PCL. Previous studies [4] have shown that there are four main mechanisms for the ROP of lactone: (1) Active hydrogen catalytic mechanism: the nucleophile attacks the carbonyl carbon of lactone to trigger the break of the acyl oxygen bond of lactone to promote the chain growth. (2) Cationic catalysis mechanism: cationic attack on the oxygen of lactone acyl oxygen bond, resulting in the break of acyl oxygen bond and ring opening. (3) Anion catalytic mechanism: similar to the catalytic mechanism of active hydrogen, anions also induce lactone ring-opening by attacking carbonyl carbon. (4) Coordination-insertion mechanism: Firstly, the metal complex and the carbonyl oxygen of the lactone produce a coordination compound through weak coordination interaction, and then through the breaking of some old coordination bonds and the formation of a new coordination bond to form a transitional metastable intermediate complex. Finally, the transition state of the complex to the stable compound promotes the transfer of coordination electrons, which leads to the break of acyl oxygen bond and ring-opening. The main reason for the catalytic performance of MOF was that MOF had a coordination unsaturated metal center. The coordination unsaturated metal center could be the metal ion of the MOF material, which was naturally coordination unsaturated, or it could be formed by heat treatment, which move away the coordinated solvent. Sattayanon and Gümüta and our preliminary study [53,54,55] demonstrated that CNF grafted with ε-CL reactions followed the coordination-insertion mechanism, the coordination-insertion mechanism of the ring opening of ε-CL by ZnEu-MOF is shown in Figure 3.

It could be seen from the reaction mechanism that the catalytic effect was not only related to Lewis acidic metal center, but also related to ligands. Although ligands do not directly participate in the reaction, they participate in nucleophilic reaction, which directly affects the grafting ratio [56]. Our previous work reported that the zinc ions can catalyze the ROP of ε-CL [22], the result of Wang et al. [57] indicated the Europium ion also had high catalytic activities towards the ROP of ε-CL.

### 3.3. Characterization of PCL and the Composite Films

#### 3.3.1. Macroscopic Mechanical Properties and Morphology

The tensile strength, elastic modulus, elongation at break, and stress–strain diagrams for all the composite films were collected, as shown in Figure 4, and Figure 4a,b shows the tensile strength (8.6 ± 0.2 MPa) and elastic modulus (73.1 ± 4.8 MPa) of pure PCL. The tensile strength of the CNF/PCL (2 wt%) composite was 6.6 ± 1.2 MPa and the elastic modulus was 63.5 ± 5.8 MPa, which were lower than pure PCL, because the CNFs easily agglomerated, making them difficult to uniformly disperse in PCL, resulting in a decrease in tensile strength and elastic modulus. The tensile strength value of CNF-S/PCL (2 wt%) was 4.3 ± 0.7 MPa and the elastic modulus was 47.0 ± 4.1 MPa. CNF-S was modified by the sulfonated method, and its tensile strength and elastic modulus were lower than pure PCL and CNF/PCL, which indicated that the sulfonation modification method did not play a role in this experiment. Thus, it could not improve the dispersion of the CNFs in the matrix PCL, and did not enhance the compatibility of the CNFs and PCL. The tensile strength of 2 wt% LCF was 9.5 ± 1.0 MPa and the elastic modulus was 71.6 ± 2.3 MPa, when the GCL was catalyzed by 4-dimethylaminopyridine (GCL-D). The tensile strength of 2 wt% LCF when GCL was catalyzed by ZnEu-MOF to form the composites (LCF) was 21.3 ± 1.0 MPa and the elastic modulus was 132.4 ± 7.9 MPa. Thus, the tensile strength and elastic modulus of LCF catalyzed by the ZnEu-MOF were higher than the LCF catalyzed by DMAP, because the MOFs promoted the dispersion of GCL in the PCL matrix. In addition, the MOF crystals were well dispersed in the PCL matrix, possibly doe the presence of existing metal-organic frames and the porous structure of the MOFs. There were two reasons for this namely, van der Waals forces such as dipole–dipole (Keesom), induced dipoles (Deby), and dispersion (London) interactions between the MOFs and the PCL molecules. Another reason was the porous structure of the MOFs, which could lead to the adsorption and diffusion of PCL molecules to the MOFs [58]. The value of LCF, where GCL was first catalyzed by the ZnEu-MOF, first increased and then decreased. The composite film with 4 wt% GCL content had the highest tensile strength and elastic modulus, with values of 23.1 ± 1.4 MPa and 133.3 ± 9.2 MPa. The tensile strength was 167.25% higher and the elastic modulus was 82.27% higher than pure PCL. When the GCL content reached 10 wt%, the tensile strength and elastic modulus of the composite film decreased to 10.9 ± 0.7 MPa and 96.9 ± 10.7 MPa, and these values were also higher than the pure PCL film. Two main reasons accounted for the improved tensile strength and elastic modulus of the composite films with GCL component doping, namely, (1) the GCL could act as a nucleating agent during the film formation process of the composite films, and (2) due to chemical compatibility with the PCL matrix, a small amount of GCL could form a continuous interface with the matrix, which was conducive to stress transfer. Based on the microstructure of the composite films observed by SEM, the reason for the decreased tensile strength and elastic modulus of the composite films was possibly because GCL was distributed in the matrix through physical bonding, and peeled off excessively when the GCL content was higher than 4 wt%. Thus, the addition of GCL could be attributed to the increase in the number of defects in the PCL composite membranes, and these poor compatibilities become stress concentration points during the tensile process, reducing the tensile strength and elastic modulus.

The elongation at break results for all composite films are shown in Figure 4c, where PCL displayed ductile fracture with an elongation at break of 729%, and when CNF and CNF-S were added, the elongation at break values were 26.97% and 51.65%, respectively. The reason was the poor compatibility between CNF, CNF-S, and the matrix PCL, with the materials not uniformly dispersed in PCL, so the elongation at break decreased sharply. The elongation at break of GCL-D/PCL was 437%, and when the content of GCL was 2 wt%, the elongation at break of LCF was 886%, which was 21.58% higher than pure PCL. In fact, low content (2 wt%, 4 wt%) of GCL addition resulted in its tendency to move with the matrix, thus, the surrounding matrix would extend when stretching. The viscous drag effect of the GCL with good adhesion to the matrix could improve the strength, elastic modulus, and elongation at break. When the GCL content reached 6 wt%, the elongation at break decreased, due to the large amount of GCL addition, and it could not uniformly disperse in the PCL matrix, resulting in a decrease in elongation at break.

Figure 4d shows the strain–stress curve of all composite films, where the addition of GCL increased the elastic modulus of PCL. As a result, this led to smaller and more uniform GCL dispersion as the stress increased during elongation. In addition, GCL could act as a bridge to prevent microcracking. Thus, enhanced interfacial interactions were observed in the PCL nanocomposites. The mechanical properties of the composite films obtained in this work were superior to those reported in the literature [12,19,23,24], as the addition of GCL significantly increased the elongation at break, tensile strength, and elastic modulus of the material simultaneously [59].

Figure 4e–j shows the micro-morphologies of the PCL and LCF composite films, where (e) presents the pure PCL film, and (f)–(j) corresponded to the composite films with GCL ratios of 2 wt%, 4 wt%, 6 wt%, 8 wt%, and 10 wt%, respectively. Each film still showed good matrix continuity, indicating that the bubbles in the solution were removed during the film formation process, with almost no large amounts of pore defects in the films. The needle-shaped and short rod-shaped particles in the composite films were the GCL components. The GCL components in the 2 wt% and 4 wt% LCF composite films were evenly distributed without large-scale aggregation, revealing good chemical compatibility between the matrix and the GCL components. With increasing GCL content, some of the GCL particles clustered and became embedded in the matrix through simple physical interactions, because the structure and surface wettability of GCL differed from that of the pure PCL matrix. This phenomenon caused the GCL particles to peel off from the matrix, leading to the formation of pits or holes, and reducing the continuity of the matrix (Figure 4h).

#### 3.3.2. Physical-Chemical Properties of PCL and the Composite Films

The XRD diffraction patterns of the CNF, GCL, ZnEu-MOF, and all composite films are shown in Figure 5a. The diffraction peaks of CNF were 2*θ* = 34.53°, 22.59°, 16.68°, 14.82° corresponded to the (004), (200), (110), and (1$\bar{1}$0) crystal planes, which were the characteristic peaks of the cellulose Iβ crystal structure [60]. In the diffraction pattern of GCL, the above peak was not observed, but a new wide peak appeared at 2*θ* = 20.78°. This peak corresponded to the (110) plane of the PCL chain [61]. The characteristic peak of CNF disappeared because the introduction of the PCL side chain was replaced the hydroxyl groups of the fibers, which weakened the hydrogen bonding of the CNFs and damaged the original crystalline structure of the CNFs. Thus, the above crystallization peaks disappeared. However, the characteristic diffraction peaks of the ZnEu-MOF were not observed, because the ZnEu-MOF content was too low. PCL showed a characteristic diffraction peak at 2*θ* = 21.50°, 22.16°, 23.78° [62], which was observed when 2–10 wt% GCL was added. The characteristic peaks of the composite films were all maintained, indicating that the addition of GCL did not change the crystal structure of PCL. In addition, the XRD diffraction peaks of the composite films were enhanced after GCL addition, compared to pure PCL, which indicated that GCL could promote the crystallization of PCL and improve the crystallinity of the composite films.

Figure 4c shows the TGA data of the pure PCL and composite membranes, where all composite films almost did not decompose under 250 °C, which indicated the superior thermostability of the PCL and composite films. However, due to the thermal degradation of PCL, significant weight loss was observed between 260 °C and 440 °C. In addition, the residual mass of the PCL and composite films were observed between 2–3 wt%. These results indicated that the introduction of GCL did not impact the thermal degradation properties of the PCL [63].

The physical properties of semi-crystalline polymers are primarily controlled by their supramolecular structure, with the thermal, mechanical, and biodegradable properties of multi-pulse polyester nanocomposites closely related to the crystallization process, making it necessary to study the crystallization behavior of polymers. The second heating scan and first cooling of the PCL and LCF composite films were investigated by DSC. The crystallization peak temperature (*T_c_*), melting temperatures (*T_m_*), heating enthalpy (Δ*H_m_*), crystallization enthalpy (Δ*H_c_*), and crystallization ($x_{c}$%) of PCL and the composite films were determined from the DSC curves (Figure 4a,b). The results are depicted in Table 2. The crystallization peak temperature of the composite films shifted to a lower temperature with increasing GCL content. Although the CNF surface was grafted with PCL side chains, this also could be attributed to the ordered nature of the molecules, as some of GCL migrated to the PCL matrix, and the growth mode of the LCF crystals consisted of heterogeneous nucleation three-dimensional growth, as in our previous work [42]. This also contributed to an increase in crystallinity, and these results were consistent with the results of Wang et al. and Zhou et al. [63,64]. The $x_{c}$% of the composite films increased with increasing GCL content, and GCL could improve the crystallinity of the composite films by promoting the middle and late stages of crystallization. However, in terms of PCL, the effect of adding different ratios GCL did not cause displacements in *T_m_*, and the *T_m_* of the PCL films did not significantly change with the addition of GCL, remaining between 54.98 °C and 56.27 °C. However, compared to the pure PCL film, composites had higher Δ*H_m_* values with the addition of GCL, because the molecules required more energy when the material changed from solid to liquid.

Biomaterials with biocompatibility will not result in an unfavorable response to the human body, with surface wettability having a greater impact, which can typically be determined by measuring the contact angle of the material [65]. The contact angles of the PCL and LCF composite films are shown in Figure 5e, where the contact angle of the pure PCL film was 76.5°, and the ratio of GCL significantly decreased the contact angle of the composite films because of the improved hydrophilicity. This could be explained by the large number of hydrophilic hydroxyl groups on the CNF main chains, even if the hydrophobic PCL became the branch of the CNFs. When the GCL ratio was 10 wt%, the contact angle of the composite film decreased to 56.5°, which was 35.40% lower than pure PCL.

#### 3.3.3. Fluorescent Properties of the Composite Films

The solid-state fluorescence results of the ZnEu-MOF, pure PCL film, and composite films are shown in Figure 6a,b. According to Figure 6a, the characteristic peaks of the ZnEu-MOF and GCL were 623 nm, under the excitation of 400 nm. which characteristic peaks could be assigned to emitting level 5D_0_ to 7F_2_ translation of Eu^3+^ ions, 5D_0_/7F_2_ transition is electric dipolar transition [33]. These results indicated that the main structure of the ZnEu-MOF did not change after it participated in the catalytic synthesis of GCL. The reaction mechanism also explained this result.

At the same time, the fluorescence of LCF composite films with different GCL contents was measured (Figure 6b). With different GCL contents, the fluorescence intensity differed from 2250 to 3550, and when the ZnEu-MOF content was 0.2 wt%, the fluorescence intensity reached This result was superior to Wang et al. [40], where the content of Eu(TTA)_3_(TPPO)_2_(TTA = α-trichloro-thiophene formyl acetone, TPPO = triphenylphosphine oxide) was 0.5 wt%, and the fluorescence intensity reached The fluorescence peaks of these films did not change at 623 nm [66,67,68]. The characteristic peaks of the excitation wavelength of the LCF composite films are shown in Table 3, where the excitation wavelength did not change, which was also at 400 nm. These results also indicated that the LCF composite films contained the main structure of the ZnEu-MOF. By comparing Figure 6a and Figure 6b, the fluorescence intensity of the composite membrane was much lower than GCL and the ZnEu-MOF. This was because the content of the ZnEu-MOF in the composite membrane was much lower than its content in GCL. Thus, the film had the ability to convert violet light into orange light.

In addition, the fluorescence mechanism of the ZnEu-MOF was studied. The theoretical UV–vis spectrum of the ZnEu-MOF was obtained by calculating the excited states of the ZnEu-MOF through TD-DFT (Figure 6c). The maximum absorption wavelength of the ZnEu-MOF was 385.89 nm, and the oscillator intensity was 0.03740, which was consistent with the excitation wavelength of 400 nm used in the fluorescence experiments. The results showed that the theoretical calculation method and selection of parameters were reasonable.

According to the calculations by the Multiwfn 3.7 program, the main orbital contribution of the maximum absorption wavelength of the ZnEu-MOF followed HOMO-1→LUMO + 12, and the orbital contribution was 84.2%. The molecular frontier orbitals related to the ZnEu-MOF are shown in Figure 6d–g. The HOMO-1 electron clouds of ZnEu-MOF were mainly distributed on the main ligand H_2_L, while the LUMO + 12 electron clouds were not only concentrated on this ligand, but also on the europium ions and carboxyl groups of the adjacent ligand. The results showed that the main orbital transition forms of the ZnEu-MOF included ligand-to-metal charge transfer (LMCT) [69] and ligand-to-ligand charge transfer (LLCT) [70].

### 3.4. In Vitro Cytotoxicity

PCL is a type of biodegradable aliphatic polymer. When used for agricultural films and drug carriers, it was necessary to investigate the biological toxicity of the composite films. The in vitro cytotoxicity activity of GCL-D/PCL and 4 wt% LCF toward HEK293T cells was investigated (Figure 7 and Appendix A). The results indicated that GCL-D/PCL was only non-toxic to the HEK293T cells at suspension concentrations of 10 µg/mL and 4 wt% LCF was 100 µg/mL after 24 h, but it could also enhance cell proliferation, because 4 wt% LCF generated relatively higher cell viability compared to GCL-D/PCL. After 48 h, the cell viability of GCL-D/PCL and 4 wt% LCF demonstrated the highest cell viability at 106% and 117%. When the 4 wt% LCF concentrations were 10 µg/mL, the cell viability gradually improved to >10.68% of GCL-D/PCL. A literature study by Zuppolini et al. [21] used the [Sn(Oct)_2_] as the catalyst to synthesize cellulose-graft-poly(ε-caprolactone). In our work, cell viability was superior to the results of Zuppolini et al. [21] after 24 h. This was attributed to the good compatibility of the LCF and the improvement in hydrophilicity (Figure 4e–j).

## 4. Conclusions

In this research, a novel dual-functional ZnEu-MOF was synthesized, which exhibited excellent properties for both fluorescence; it was used to catalyze wood cellulose nanofibers (CNF) and poly(ε-caprolactone) (ε-CL) formed wood cellulose nanofibers grafted with ε-CL (GCL) with a grafting ratio of 79.6%. Further, the GCL products embedded with ZnEu-MOF can function as carrier of fluorescent additive and bridge between ZnEu-MOF and PCL matrix. It is noteworthy that the whole framework of the embedded ZnEu-MOF in the final light composite films (LCFs), the excitation wavelength (*E*_x_ = 400 nm), the emission wavelength of the GCL and composite film was also 623 nm, which was generally consistent with that of ZnEu-MOF. It still maintained its integrity that guarantees practical application. More interesting, the composite film with 4 wt% GCL content had the highest tensile strength and elastic modulus of 23.1 ± 1.4 MPa and 133.3 ± 9.2 MPa. The tensile strength was 167.25% higher and elastic modulus was 82.27% higher than that pure PCL, When the content of GCL reached 10 wt%, the tensile strength, elastic modulus of the composite film decreased to 10.9 ± 0.7 MPa and 96.9 ± 10.7 MPa, which was also higher than that of the pure PCL film. The addition of GCL significantly increased the elongation at break, tensile strength and elastic modulus of the material. SEM revealing the interfacial interactions between the CNF and matrix were improved after the graft modification by ZnEu-MOF as catalyst. The addition of GCL improved the hydrophilicity of PCL. The results of DSC and TG showed that LCF has good thermal stability. Cytotoxicity test showed that 4 wt% LCF was non-toxic to the HEK293T cells but also promoted HEK293T cells growth within a certain concentration range. Our research is expected to shed light on the fabrication of functional MOFs for practical applications by introducing mixed metals in the nodes. ZnEu-MOF represents the novel bifunctional material for ROP reactions and light composite films, respectively.

## Figures and Tables

**Figure 1 nanomaterials-13-01904-f001:**
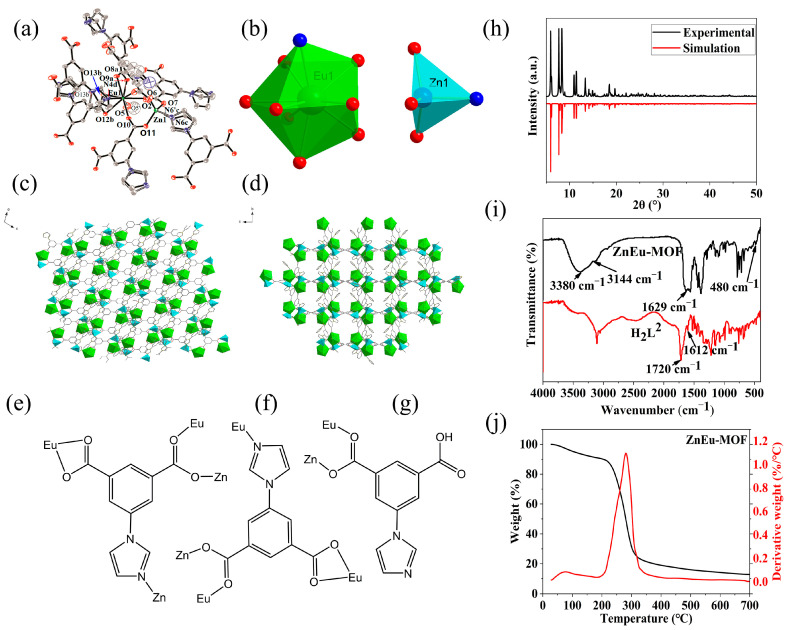
(**a**) Unit cell and coordination environment of ZnEu-MOF, solvent water molecules were omitted. (**b**) Coordination modes of Zn1 and Eu1. (**c**) Staking diagram of ZnEu-MOF along b axis. (**d**) Staking diagram of ZnEu-MOF along an axis. (**e**–**g**) Coordination modes of H_2_L. (**h**) Powder XRD pattern of ZnEu-MOF and the simulated pattern. Infrared spectrum of ZnEu-MOF and H_2_L ligand. (**i**) Infrared spectrum of ZnEu-MOF and H_2_L ligand. (**j**) The TG and DTG curves of ZnEu-MOF.

**Figure 2 nanomaterials-13-01904-f002:**
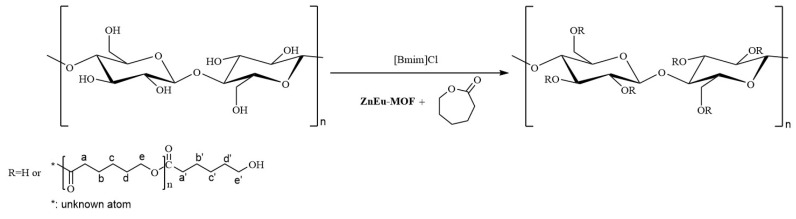
Synthesis process of graft copolymer (GCL).

**Figure 3 nanomaterials-13-01904-f003:**
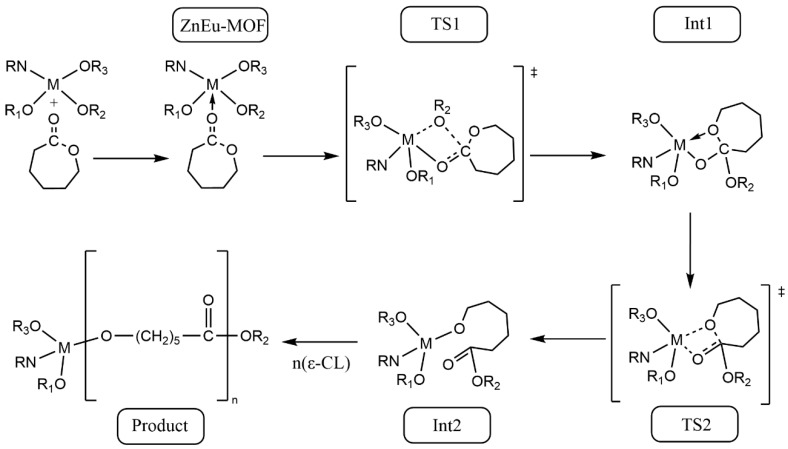
Coordination insertion mechanism of ROP of ε-caprolactone (‡: Transition state).

**Figure 4 nanomaterials-13-01904-f004:**
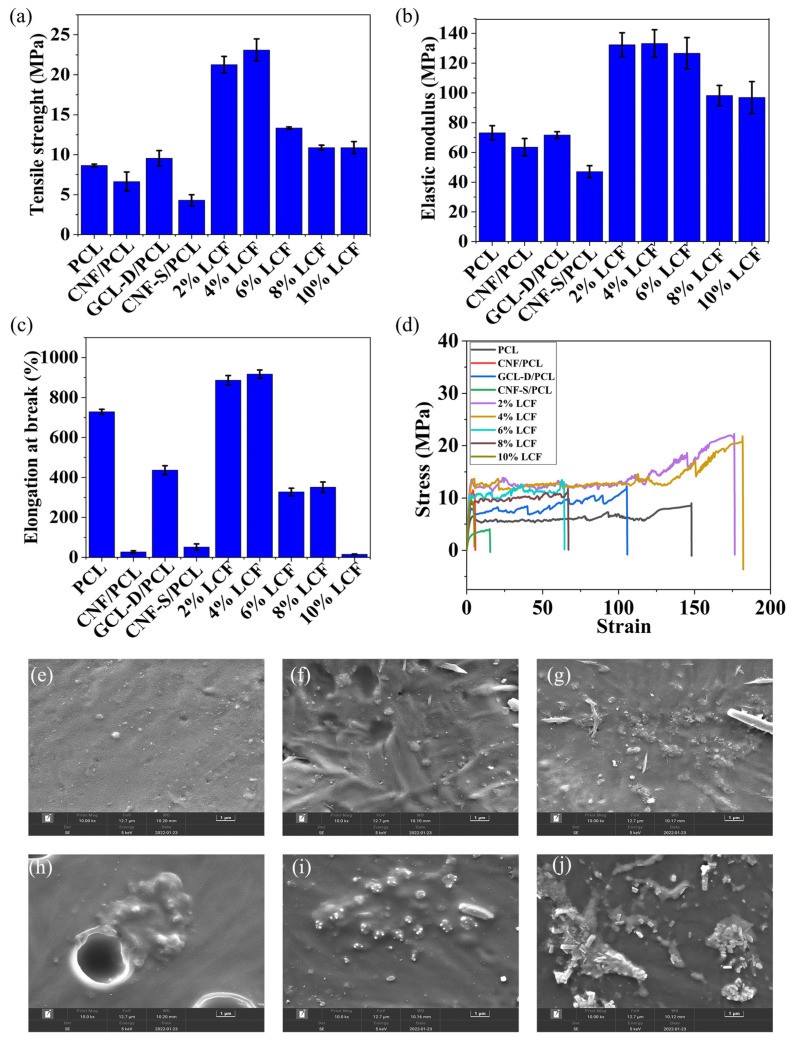
Macroscopic mechanical properties, morphology of the pure PCL and composite films: (**a**) tensile strength, (**b**) elastic modulus, (**c**) elongation at break and (**d**) stress–strain diagrams of pure PCL film and composite films; (**e**–**j**) micromorphology of (**e**) pure PCL and LCF composite films with different GCL ratios: (**f**) 2 wt%, (**g**) 4 wt%, (**h**) 6 wt%, (**i**) 8 wt%, and (**j**) 10 wt%.

**Figure 5 nanomaterials-13-01904-f005:**
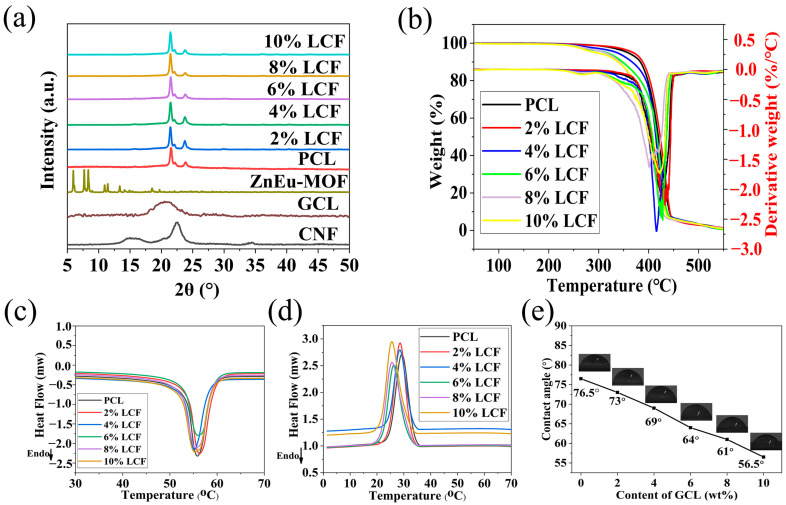
Physical-chemical properties of PCL and composite films: (**a**) powder XRD pattern; (**b**) TG and DTG curves; (**c**) DSC curves of secondary heating; (**d**) DSC curves of primary cooling; (**e**) contact angles.

**Figure 6 nanomaterials-13-01904-f006:**
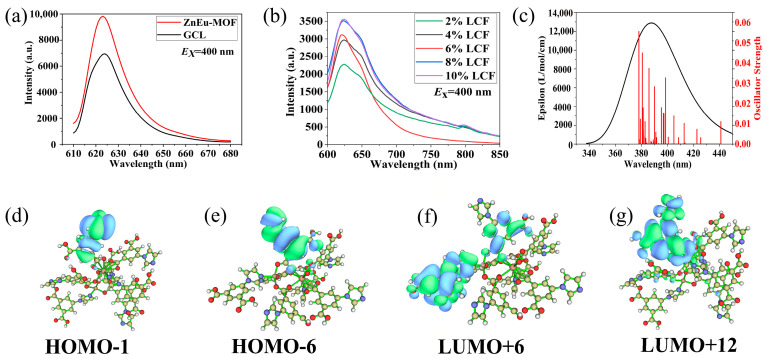
Fluorescent properties of composite films: (**a**) fluorescence of ZnEu-MOF and GCL; (**b**) fluorescence of composite films; (**c**) the calculated UV–vis spectrum of ZnEu-MOF; (**d**–**g**) molecular frontier orbitals of ZnEu-MOF.

**Figure 7 nanomaterials-13-01904-f007:**
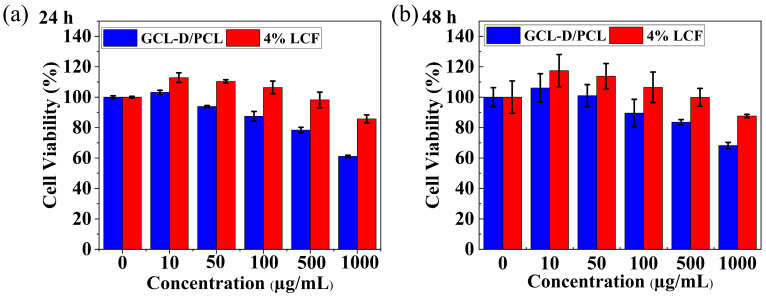
Cytotoxicity studies of GCL-D/PCL and 4 wt% LCF to HEK293T cells. (**a**) Cell viability at 24 h after treatment; (**b**) Cell viability 48 h after treatment.

**Table 1 nanomaterials-13-01904-t001:** Comparison of the catalytic activity of catalyst.

Sample	[AGU]/[ε-CL]	Catalyst (wt%)	*T* (°C)	*MS* ^a^	*W_PCL_* (%) ^b^	Ref.
CGCL ^c^	1:10	2% Sn(Oct)_2_	130	0.68	32.36	[20]
BGCL ^c^	1:50	2% Zn-MOF	120	5.72	80.10	[22]
GCL ^c^	1:10	2% Sn(Oct)_2_	120	1.32	48.07	[41]
GCL ^c^	1:10	2% DMAP	120	0.42	22.48	[41]
GCL ^c^	1:10	2% UiO-67	120	1.45	50.41	[41]
GCL2	1:10	2% MOF 2	120	1.09	43.52	[41]
GCL1	1:10	2% MOF 1	120	4.38	75.52	[41]
GCL1	1:30	2% MOF 1	120	9.56	87.05	[41]
GCL	1:30	2% ZnEu-MOF	120	5.47	79.38	This work

^a^ Molar composition in graft copolymer as calculated via ^1^H NMR spectroscopy: *MS* = (*I_a_* + *I_a′_*)/2*H*_4_. ^b^ PCL content, as calculated via ^1^H NMR spectroscopy: *W_PCL_* = 100*MS* × 114/(162 + *MS ×* 114). ^c^ CGCL: cellulose-g-poly(ε-caprolactone), BGCL: banana cellulose-g-poly(ε-caprolactone), and GCL: wood nanofiber-g-poly(ε-caprolactone); MOF **1**: [Zn_2_(L)_2_(HIPA)]_n_ (1); MOF **2**: [Zn_9_(L)_6_(BTEC)_3_(H_2_O)_4_·6H_2_O]_n_; (2) (HL = 3-amino-1H-1,2,4-triazole, H_2_HIPA = 5-hydroxyisopht-halic acid, H_4_BTEC = benzene-1,2,4,5-tetracarboxylic acid).

**Table 2 nanomaterials-13-01904-t002:** The DSC data of PCL and LCF films at the second heating scan.

Sample	*T_c_* (°C)	*T_m_* (°C)	Δ*H_m_* (J/g)	Δ*H_c_* (J/g)	$x_{c}\%$
PCL	29.28	55.82	52.08	56.31	38.29%
2 wt% LCF	28.59	56.27	52.98	59.35	39.75%
4 wt% LCF	28.53	55.16	52.63	46.90	40.31%
6 wt% LCF	26.28	55.99	53.36	50.37	41.74%
8 wt% LCF	26.02	54.98	56.14	57.58	44.87%
10 wt% LCF	25.61	55.50	58.86	59.47	47.04%

**Table 3 nanomaterials-13-01904-t003:** Characteristic peaks of excitation wavelength of PCL and LCF composite films.

Film Number	PCL	2 wt% LCF	4 wt% LCF	6 wt% LCF	8 wt% LCF	10 wt% LCF
Excitation wavelength	-	400.6 nm	403.2 nm	401.0 nm	397.0 nm	402.8 nm

## Data Availability

Data will be made available on request.

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
