# Peer review of "Wood Cellulose Nanofibers Grafted with Poly(ε-caprolactone) Catalyzed by ZnEu-MOF for Functionalization and Surface Modification of PCL Films"

_nanomaterials, 2023, doi:10.3390/nano13131904_

Round 1
Reviewer 1 Report
The article from Guo and coworkers represents the description of a ZnEu-based MOF, with improvements with respect to previous studies, for what concerns mechanical and catalytic properties. The design of the MOF aims to favor the grafting of renewable cellulose nanofibers reinforced with polycaprolactone, which are materials of high interest for many industrial use, spanning from the food packaging to drug release.
Authors have provided an accurate characterization of the MOF, as well as of its improved properties. The article is, generally, well written, with results clearly presented and coherently discussed. For these reasons, I repute that it can be of potential interest for the journal readership community.
I therefore recommend the publication in Nanomaterials after the consideration of the following minor revisions:
-I understand and appreciate the will of the authors in providing a comprehensive background of the topics, but I found that the introduction, in some sections, is too long. For instance, authors could find a way to reduce the part describing the previous synthetic efforts of other researchers, i.e. from "Habibi et al" at line 18 to line 37 of page 2;
-I did not find any information about the computational methods adopted to carry out the TD-DFT methods. Authors should include it in the materials section or, at least, refer to computational protocol used in already published article;
-As a chemist, I would be interested in some hypothesis or evidence about the catalytic mechanism. I found some of this evaluable information in the supplementary material; why don't the authors put it in the main text? My impression is that the entire story would be complete and that the manuscript would increase in readability;
-Linked to the previous comment, authors should discuss or hypothesize, if available, if any catalytic role is owned by the Eu(III), in comparison with similar systems adopted for the reaction. In my opinion, this would catch the attention of theoretical and experimental equipes interested in the field.
Author Response
Response to the comments
Reviewer 1
The article from Guo and coworkers represents the description of a ZnEu-based MOF, with improvements with respect to previous studies, for what concerns mechanical and catalytic properties. The design of the MOF aims to favor the grafting of renewable cellulose nanofibers reinforced with polycaprolactone, which are materials of high interest for many industrial use, spanning from the food packaging to drug release.
Authors have provided an accurate characterization of the MOF, as well as of its improved properties. The article is, generally, well written, with results clearly presented and coherently discussed. For these reasons, I repute that it can be of potential interest for the journal readership community.
I therefore recommend the publication in Nanomaterials after the consideration of the following minor revisions:
1.Question: I understand and appreciate the will of the authors in providing a comprehensive background of the topics, but I found that the introduction, in some sections, is too long. For instance, authors could find a way to reduce the part describing the previous synthetic efforts of other researchers, i.e. from "Habibi et al" at line 18 to line 37 of page 2.
Response: Thank you so much for this valuable comment. We accepted the reviewer’s advice. The changes are highlighted in red font on the revised manuscript (Line 18-25 Page 2).
- Question:I did not find any information about the computational methods adopted to carry out the TD-DFT methods. Authors should include it in the materials section or, at least, refer to computational protocol used in already published article.
Response: Thank you so much for this valuable comment. We accepted the reviewer’s advice. We have written the TD-DFT methods in 2.5. Characterization (Line 4-11 Page 5), the changes are highlighted in red font on the revised manuscript.
- Question: As a chemist, I would be interested in some hypothesis or evidence about the catalytic mechanism. I found some of this evaluable information in the supplementary material; why don't the authors put it in the main text? My impression is that the entire story would be complete and that the manuscript would increase in readability.
Response: Thank you for your positive suggestions, the changes are highlighted in red font on the revised manuscript (Line 17-44 Page 7 ; Figure 3 Page).
- Question: Linked to the previous comment, authors should discuss or hypothesize, if available, if any catalytic role is owned by the Eu(III), in comparison with similar systems adopted for the reaction. In my opinion, this would catch the attention of theoretical and experimental equipes interested in the field.
Response: Special thanks to you for your positive comments. We accepted the reviewer’s advice and the alterations are highlighted in red font on the revised manuscript (Line 39-44 Page 7) and References [22],[56-57].
Reviewer 2 Report
In this paper, the authors show a renewable cellulose nanofiber (CNF) reinforced biodegradable polymers, using a novel ZnEu-MOF as the catalyst graft copolymers (CGP) with CNFs. Mechanical and fluorescent properties were obtained in the LCFs.
This paper in interesting and well documented. The references are complete. The Tables and Figures are very good. This paper can be published in Nanomaterials but is necessary that the authors reviewer the manuscript.
Comment
1) Abstract: define the letters PCL, ROP….Revise all manuscript
2) Introduction: change et al.[23]grafted for et al. [23] grafted
3) Introduction: Revise the paragraph “For example, well documented…etc. [36-41]. The formulas are confused
4) Experimental Part: Change (0.4 mmol, 0.0929 g) for (0.40 mmol, 0.093 g). Revise all manuscript
5) Experimental Part: Change yield 0.0778 g 82.38% for 0.0778 g for 82.4%
6) Revise the description of the structure ZnEu-MOF. The description is confuse. Revise this Part
7) Change TG for TG/DTG
8) Reordered the characterization, first XRD
9) Write the difference between ZnEu-MOF described in this paper and other ZnEu-MOF described previously in the literature [22, 43]
10) Page 8, revise the values 728.72%, 436.92%, 885.92%
11) Revise the Part 3.3.3. Fluorescent Properties
Is necessary that the authors reveiwer the English
Author Response
Response to the comments
Reviewer 2
In this paper, the authors show a renewable cellulose nanofiber (CNF) reinforced biodegradable polymers, using a novel ZnEu-MOF as the catalyst graft copolymers (CGP) with CNFs. Mechanical and fluorescent properties were obtained in the LCFs.
This paper in interesting and well documented. The references are complete. The Tables and Figures are very good. This paper can be published in Nanomaterials but is necessary that the authors reviewer the manuscript.
- Question: Abstract: define the letters PCL, ROP, Revise all manuscript.
Response: Thank you for your positive suggestion. According to the reviewer’s suggestion, the changes are highlighted in red font on the revised manuscript and SI.
- Question:Introduction: change et al.[23] grafted for et al. [23] grafted
Response: Thank you for your positive suggestions, the changes are highlighted in red font on the revised manuscript (Line 18 Page 2).
- Question: Introduction: Revise the paragraph “For example, well documented…etc. [36-41]. The formulas are confused.
Response: Thank you for your positive suggestion. According to the reviewer’s suggestion, the changes are highlighted in red font on the revised manuscript (Line 50 Page 2).
- Question: Experimental Part: Change (0.4 mmol, 0.0929 g) for (0.40 mmol, 0.093 g). Revise all manuscript
Response: Thank you for your positive suggestion. According to the reviewer’s suggestion, the changes are highlighted in red font on the revised manuscript (Line 32-33 Page 3).
- Question:Experimental Part: Change yield 0.0778 g 82.38% for 0.0778 g for 82.4%
Response: Thank you for your positive suggestion. According to the reviewer’s suggestion, the changes are highlighted in red font on the revised manuscript (Line 38 Page 3).
- Question:Revise the description of the structure ZnEu-MOF. The description is confuse. Revise this Part
Response: Thank you for your positive suggestion. According to the reviewer’s suggestion, the changes are highlighted in red font on the revised manuscript.
- Question: Change TG for TG/DTG
Response: Thank you for your positive suggestion. According to the reviewer’s suggestion, the changes are highlighted in red font on the revised manuscript (Figure 5b Page 8).
- Question: Reordered the characterization, first XRD
Response: Thank you for your positive suggestion. According to the reviewer’s suggestion, the changes are highlighted in red font on the revised manuscript (Figure 5a Page 8).
- Question: Write the difference between ZnEu-MOF described in this paper and other ZnEu-MOF described previously in the literature [22, 43].
Response: Thank you for your positive suggestion. In the literature [22, 43], we synthesized mononuclear Zn-MOF, in this paper, we synthesized heteronuclear MOF containing both transition metal Zn and rare earth metal Eu.
- Question:Page 8, revise the values 728.72%, 436.92%, 885.92%
Response: Thank you for your positive suggestion. According to the reviewer’s suggestion, the changes are highlighted in red font on the revised manuscript.
- Question: Revise the Part 3.3.3. Fluorescent Properties
Response: Thank you for your positive suggestion. According to the reviewer’s suggestion, the changes are highlighted in red font on the revised manuscript (Line 19-20 Page 12).
Reviewer 3 Report
The presented work undoubtedly corresponds to the current trends in the field of creation and application of MOF and polymeric materials based on them. The authors use the necessary set of methods to analyze the obtained MOF, as well as the obtained composite films.
After reading the paper, some comments arose:
1. The authors indicate that after TGA, a mixture of Zn and Eu oxides is formed in the residue. Was there an XRD analysis of the resulting residue? The cited link indicates the analysis of zinc-containing MOF.
2. In the IR spectrum of coordination polymer, there are no deformation vibrations of water molecules, which indicates the absence of free or crystallization water in the lattice. Was the MOF additionally treated prior to the polymerization process? Was the water removed? Since water greatly reduces the efficiency of the process when producing polycaprolactones on tin catalysts.
3. The degree of polymerization and modification of cellulose nanofibers was evaluated by NMR. However, this analysis is very inaccurate, as shown (supporting materials). The spectrum should be given in a better resolution and with a clearer correlation. The spectrum is dominated by solvent proton signals.
After corrections are made, the manuscript could be published in Nanomaterials.
Author Response
Response to the comments
Reviewer 3
The presented work undoubtedly corresponds to the current trends in the field of creation and application of MOF and polymeric materials based on them. The authors use the necessary set of methods to analyze the obtained MOF, as well as the obtained composite films.
After reading the paper, some comments arose:
1.Question: The authors indicate that after TGA, a mixture of Zn and Eu oxides is formed in the residue. Was there an XRD analysis of the resulting residue? The cited link indicates the analysis of zinc-containing MOF.
Response: Thank you for your positive suggestion. We accepted the referee’s advice and the statements have been added in the manuscript and Supporting Information. The alterations are highlighted in red font on the revised manuscript(Line 20-21 Page 6)and Fig. S1 of Supporting Information.
- Question: In the IR spectrum of coordination polymer, there are no deformation vibrations of water molecules, which indicates the absence of free or crystallization water in the lattice. Was the MOF additionally treated prior to the polymerization process? Was the water removed? Since water greatly reduces the efficiency of the process when producing polycaprolactoneson tin catalysts.
Response: Thank you for your positive suggestion. The water of MOF was removed by heated in a tube furnace at 200 ◦C for 14 h under vacuum (0.09 MPa), the graft copolymer was synthesized under anaerobic and anhydrous conditions. Water also greatly reduces the efficiency of the process when producing graft copolymer on MOF catalysts. The alterations are highlighted in red font on the revised manuscript(Line 27-28 Page 3)
- Question: The degree of polymerization and modification of cellulose nanofibers was evaluated by NMR. However, this analysis is very inaccurate, as shown (supporting materials). The spectrum should be given in a better resolution and with a clearer correlation.The spectrum is dominated by solvent proton signals.
Response: Thank you for your positive suggestion. We have retested the experiments to reduce the solvent proton signals, offer a better resolution and with a clearer correlation, the results are showed in Fig S2a of supporting information.
The reviewer put forward that the degree of polymerization and modification of cellulose nanofibers was evaluated by NMR. However, this analysis is very inaccurate. But many researchers had used NMR to obtain the degree of polymerization. For example: â‘ Guo, Y.; Wang, X.; Shen, Z.; Shu, X.; Sun, R. Preparation of Cellulose-Graft-Poly(ε-caprolactone) Nanomicelles by Homogeneous Rop in Ionic Liquid. Carbohydr. Polym. 2013, 921, 77-83. â‘¡Zuppolini, S.; Maya, I. C.; Diodato, L.; Guarino, V.; Borriello, A.; Ambrosio, L. Self-Associating Cellu-lose-Graft-Poly(ε-caprolactone) to Design Nanoparticles for Drug Release. Mater. Sci. Eng. C 2020, 108, 110385. â‘¢Herrera, N.; Olsén, P.; Berglund, L. Strongly Improved Mechanical Properties of Thermoplastic Biocomposites by PCL-Grafting inside Holocellulose Wood Fibers. ACS Sustain. Chem. Eng. 2020, 8, 11977-11985.â‘£ Jiang, C.; Wang, X.; Sun, P.; Yang, C. Synthesis and Solution Behavior of Poly(ε-Caprolactone) Grafted Hydroxyethyl Cellulose Copolymers. Int. J. Biol. Macromol. 2011, 48, 210-214.and so on. Because the GCL is difficult to dissolve in organic solvents, so the signals of spectrum were weak, it was consistent with the literature reported. In this work, we retested the experiments, the grafting ratio of GCL was 0.2% higher than previous experiment. So we think this method is feasible to obtain the the degree of polymerization.